# Using Ecological Modelling to Assess the Long-Term Survival of the West-Indian Manatee (*Trichechus manatus*) in the Panama Canal

**Giselle Muschett** [1,*] **and Narkis S. Morales** [2,3,*] 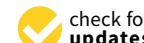

[1]  Instituto de Biología, Facultad de Ciencias, Pontificia Universidad Católica de Valparaíso, Curauma, Valparaiso 2373223, Chile

[2]  Centro de Modelación y Monitoreo de Ecosistemas, Facultad de Ciencias, Universidad Mayor, Santiago 8340589, Chile

[3]  Escuela de Ingeniería Forestal, Facultad de Ciencias, Universidad Mayor, Santiago 8580745, Chile

\*  Correspondence: giselle.muschett@pucv.cl (G.M.); narkis.morales@umayor.cl (N.S.M.)

**Abstract:** There is evidence of a thriving population of West-Indian manatee (*Trichechus manatus*) in the Panama Canal, although it is not clear if they are descendants of a small group of introduced manatees or if manatees have entered the Canal from the Caribbean through the Canal locks. This study describes the development and application of an individual-based model to assess the survival of a population of West-Indian manatees in the Panama Canal. In addition, we seek to determine the effects of isolation, predation, and mortality on long term survival. The model was parameterized using empirical data collected from the literature to every extent possible. A sensitivity analysis was performed to evaluate the model's sensitivity to changes in the used parameters. Four scenarios were modeled to understand under which conditions the original population could have been maintained over time. Our results show that the manatee population would have collapsed quite quickly after its initial introduction and that only through the addition of several individuals into the lake over the years could the population have survived until the present day. Our results have important implications for the long-term conservation of this endangered species.

**Keywords:** introduced species; Lake Gatun; conservation; manatee

## 1. Introduction

The West-Indian manatee (*Trichechus manatus*) is the most broadly distributed of all manatees, but despite its expansive range, it is a highly endangered species [1,2]. Manatees are typically found in slow-moving rivers, coastal lagoons, and estuaries along the Atlantic and Caribbean coast from the southern United States, through Central America to Northern Brazil, including the Antilles. These habitats, however, are typically highly sought after for coastal real estate developments, which brings manatees into direct competition with humans [1,2]. In addition to habitat loss and degradation from coastal development, manatees are threatened by pollution, collisions with watercraft, and hunting throughout its distribution [3,4]. As a consecuence, most manatee populations are small and have limited interconnectivity [1,2]. Outside of the United States, reliable information on manatee numbers and conservation status is scarce. Although there have been some notable recent attempts [4,5], due in large part to the high costs involved in manatee research, most countries have limited data and rely on anecdotal evidence to estimate manatee populations or assess long term survival.

Panama hosts several small manatee populations, but the two largest are located in (1) Bocas del Toro, and (2) Lake Gatun, the main source of water for the operation of the Panama Canal [1,6] (Figure 1). Both of these populations are under heavy pressure from human activities. The Bocas

del Toro population was recently estimated at between 22 and 70 individuals [6]. While hunting manatees is outlawed in Panama [7], sustenance hunting is still carried out occasionally by local fishermen [8]. The Bocas del Toro manatees also suffer from habitat loss and degradation through human development and pollution of local waterways from agricultural runoff and sewage, as well as collision with watercraft [9]. In contrast, the population in Lake Gatun is much smaller and estimated to be around 30 individuals [10]. The main threats to manatees in Gatun are collisions with transoceanic vessels and death as a result of dredging operations for the Panama Canal. Hunting seems much less of a threat for this population mainly because the population density is so low, and because hunting and fishing activities in the Canal watershed are heavily regulated by the Panama Canal Authority (ACP) [10]. This latter population provides a unique study system among manatees due to the economic importance of the Panama Canal, and the particularity of Lake Gatun as a potentially closed system.

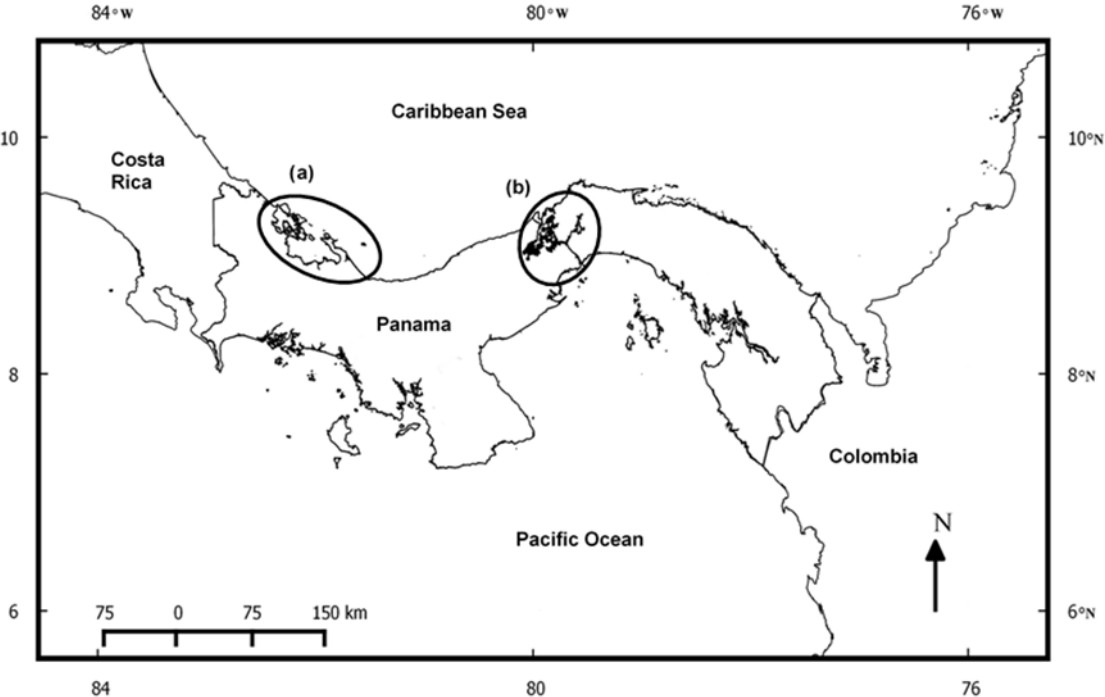

**Figure 1.** The two largest populations of West Indian manatees (*Trichechus manatus*) in Panama; (**a**) Bocas del Toro and (**b**) Lake Gatun, the main body of water for the operation of the Panama Canal. There are also occasional manatee sightings along the Caribbean coast of Panama.

Gatun is an artificial lake created during the construction of the Panama Canal between 1907 and 1913 by the damning of the Chagres River. Currently, close to 14,000 vessels use the Canal each year, with an average of 38 vessels per day. Each crossing requires almost 200 million liters of water, and while the new locks allow for water conservation, most of that freshwater is lost to the sea [11]. Despite its anthropogenic origins, Lake Gatun offers many ecosystem services. The lake is the main source of freshwater for Panama and Colon cities, and also generates hydroelectric power for Panama City [12]. There are many local communities that depend on the lake for sustenance fishing, and several commercial fishing companies also make a living from the lake. Gatun also provides recreational opportunities for local communities. During the creation of the lake, as the valley flooded, terrestrial wildlife sought the relative safety of higher ground. Hilltops became islands within the lake, and, in 1923, the largest of these was set aside as a nature reserve named Barro Colorado Island (BCI), administered by the Smithsonian Tropical Research Institute (STRI) [13]. BCI is a living laboratory open to researchers from around the world.

Currently, there is some debate as to whether the manatees in Lake Gatun are native or were introduced. While there is anecdotal evidence to suggest manatees existed in the Chagres river prior to the creation of this lake, swimming freely between the Chagres river and the Caribbean, it is uncertain whether those manatees would have survived the 10-year construction process of the Panama Canal [14,15]. Indeed, there are no known sightings of manatees in the Lake prior to the 1960s. In 1964, the then Panama Canal Commission introduced ten manatees into the Lake as part of an aquatic vegetation control program: one Amazonian manatee *T. inunguis* from Peru and nine West-Indian manatees *T. manatus* from Bocas del Toro. The *T. inunguis* that was introduced was male, and there were six male and three female *T. manatus*. There is no data on the age of the individuals, but they were presumably all adults. These 10 manatees were translocated into an enclosure in Lake Gatun to study their effectiveness at vegetation control [14]. After a monitoring period of two years, it became evident that the manatees had developed selective eating habits due to the abundance and variety of vegetation available [14]. Shortly after this time, the manatees either escaped or were released into the lake, and the aquatic vegetation control program was abandoned [14]. Since then, manatee sightings in the lake have been relatively common [10,15]. However, there is an obvious conflict between the economic interests of the Panama Canal and the conservation needs of these manatees.

Knowledge of manatee life history would suggest that a small founding population of only 10 individuals would not fare well in the long term. Low birth rates [16,17], disease [18], inbreeding depression, and marked founder effects [19] would lead to a rapid decline in manatee numbers in Lake Gatun after their translocation. With climate change already affecting the lake [20], the available habitat for manatees will be reduced, forcing them ever closer to the dredged (i.e., deeper) sections of the lake where they will be under direct threat from large oceanic vessels and further dredging activities [21,22]. The recent expansion of the Panama Canal, with larger locks that allow the transit of Neo-Panamax vessels, these manatees face a frequency and scale of aquatic transit unlike any other population in the world. A recent detailed review of the Panama Canal Aquatic Vegetation Control Unit logbooks and Smithsonian Tropical Research Institute (STRI) Game Warden reports revealed 17 manatee carcasses recovered from the waterway between 1995 and 2007 [10]. Though no necropsies were performed, based on injuries and timing of carcass sightings, due to the constant dredging activities (which include underwater detonations), Panama Canal personnel attributed these deaths to either the dredging activities, or collisions with the large transoceanic vessels that transit the lake [10]. It is likely that this mortality rate will increase with the increased traffic of larger vessels, coupled with a reduction in lake surface area due to higher temperatures and decreased rainfall brought about by climate change.

Despite these threats, there numerous sightings of manatees in the Lake [10,14,15], during an aerial survey in 2007, Muschett and Vianna [10] sighted 16 manatees, including females with newborn calves and juveniles. Considering that aerial surveys in murky tropical waters tend to underestimate true abundance [23,24], the manatee population in Gatun could be estimated at some 30 individuals and would be of great importance to Panama. The above evidence begs the question of how this small and apparently isolated manatee population has survived for over 50 years. While there have been numerous possible scenarios proposed, one popular hypothesis suggests that there is a low but relatively frequent influx of manatees from the Caribbean Sea into Lake Gatun via the Gatun locks (Figure 2). Indeed, a possible long-term conservation strategy for the manatees in the lake would be to carry out periodic and strategic introductions of manatees from the Caribbean coast. Nevertheless, such introductions could only be carried out as a result of long-term monitoring and conservation program, where clear priorities and methodologies were established. To date, there is no evidence on whether manatees can pass through the Canal locks unaided.

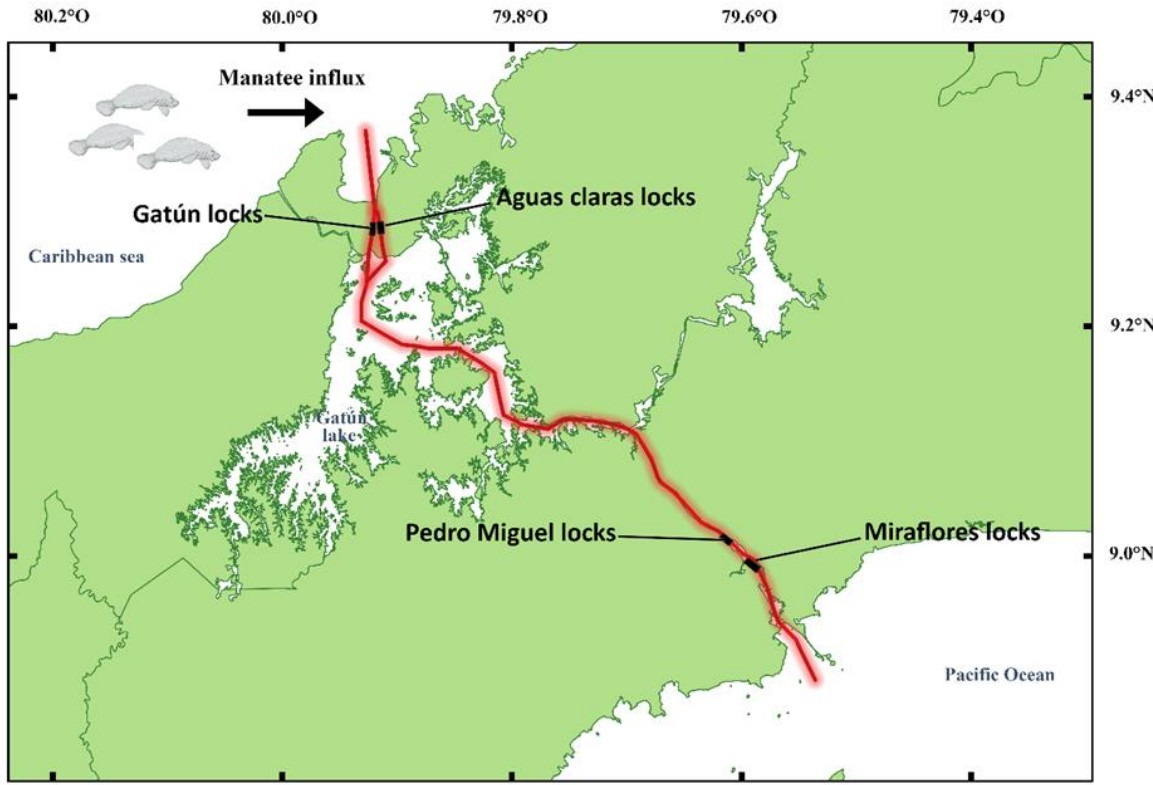

**Figure 2.** Image of lake Gatun highlighting the manatee influx from the Caribbean Sea into Lake Gatun through the Panama Canal locks. The red line indicates the navigation route for ships traversing the Panama Canal.

A cost-effective method to determine whether manatees are passing through the locks into the Caribbean Sea (and vice-versa) would go a long way to understanding how this population has survived and would provide much-needed information for the development of recommendations and a comprehensive management program for the lake. A mark and recapture program using radio tags [23], or a long-term monitoring program, be it through aerial surveys or the use of radar technology [6,23,25], would help assess whether manatees in Lake Gatun are indeed isolated. However, these types of monitoring programs are time-consuming and expensive, and in the case of Lake Gatun, would require permits from the Panama Canal Authority. A low-cost alternative would be the use of an ecological model. Individual-based models (IBMs) have become a popular tool, allowing population viability assessments of different species under a wide variety of scenarios [26–28]. By first developing a baseline and the altering factors or conditions, we can develop scenarios that would be impractical or near impossible to replicate in the field. A posteriori analysis of the model's results allows researchers to further improve and validate the model. For example, Liu et al. [26] modeled the locations and movement patterns of the wood mouse (*Apodemus sylvaticus*) to evaluate the potential exposure to agricultural pesticides. The authors validated the model, contrasting their results with data from field experiments.

In this study, we used an ecological model to assess the effects of isolation, predation, and mortality on the long-term survival of the population of *T. manatus* in the Panama Canal. Our aim was to determine whether this manatee population would survive more than 50 years based solely on the original translocated population to demonstrate how vulnerable this population is to the current and future anthropogenic pressures within its habitat. To achieve this goal, we developed an individual-based model that included biological, anthropogenic, and environmental variables to determine the long-term viability of the manatee population in Lake Gatun since its introduction in 1964. The results of our study have implications for the long-term conservation and survival of this unique manatee population.

We also make a series of recommendations to ensure that this population of manatees survives in the long term. The Panama Canal, renowned for its economic importance, can be a great ally in the maintenance of a healthy aquatic ecosystem and the conservation of this endangered species.

## 2. Materials and Methods

### 2.1. Study Species

The West-Indian manatee (*T. manatus*) is a member of the Sirenia, the only order of marine mammals that are obligate herbivores [29]. Adult manatees can measure between 2.5 and 4m in length and weigh up to 800kg [30,31]. Manatees prefer slow-moving rivers, estuaries, and coastal habitats with abundant vegetation. There are currently two major manatee populations in Panama: one in Bocas del Toro and the other in Lake Gatun in the Panama Canal [15]; Figure 1). Lake Gatun is a shallow artificial freshwater lake created for the construction of the Panama Canal. It has an area of 425 km$^2$, with a maximum depth of 30 m [11].

### 2.2. Model Description

The model corresponds to an individual-based model (IBM) designed to represent the population dynamics of the West Indian manatee (*T. manatus*) in Gatun Lake, the main source of water for the operation of the Panama Canal. The model was designed to shed light on whether the manatees in the Lake could survive based solely on the individuals translocated into Gatun Lake in the 1960s. The model was mostly empirically calibrated using available literature, reports, as well as information from the captains' logs of the Panama Canal Authority Aquatic Vegetation Control Unit and the Smithsonian Tropical Research Institute (STRI) Game Warden reports. However, there were some parameters that needed manual calibration. The model was developed using the Netlogo language programming environment (Netlogo 6.0.4; [32]). The information presented in this section is a summary of the Overview, Design, Concepts, and Details (ODD) protocol proposed by Grimm et al. [33] (Appendix A).

The model includes only one manatee species. Each individual is categorized into two reproductive stages (adult and offspring) and sex (male or female). Each tickor model run represents a year; during this time-step, several population processes take place in the following order: movement, old-age mortality, reproduction, offspring mortality, female mortality, male mortality, mortality by ships, mortality by underwater dredging detonations, offspring growth, the influx of manatees from outside the lake, and the estimated carrying-capacity of the lake (Figure 3).

To simulate the artificial introduction of manatees into Lake Gatun, the model is initialized with six adult male and three adult female manatees based on MacLaren [14]. While there are reports of hybridization between *T. inunguis* and *T. manatus* [34], we did not consider hybrid offspring in this study mainly due to the unlikelyhood of mating opportunities. During mating, manatees form mating herds of up to 20 males that jostle for position around a female [29,35,36]. Because *T. inunguis* are smaller than *T. manatus* [36], it is unlikely the single *T. inunguis* male would be able to compete and successfully mate with a *T. manatus* female. Furthermore, there are no known studies on the life history traits of manatee hybrids (e.g., calving interval, gestation period), and thus it would not be possible to include them in the model.

The individual manatees are randomly distributed in the simulation area ("world"). The initial parametrization is based on the existing literature and reports from different sources (e.g., mortality by ship strike, reproductive rate). At the initialization stage, abundances for males (six individuals) and females (three individuals) are fixed, although these can be adjusted. Age also is fixed to six years for all the simulated individuals at this early stage. Also, only one offspring was simulated at this stage; however, a slider allows us to add any number of individuals. The last breeding event refers to the latest or most recent breeding cycle and is set randomly between one and four for each female. After the initialization, there is an additional arrival of an adjustable number of adult manatees of random sex from outside the lake per tick (manatee-influx). All the individuals move randomly across the lake

("world"). There are four processes that affect mortality: carrying capacity, senescence, passing ships, mortality by detonations, and base mortality (Appendix A).

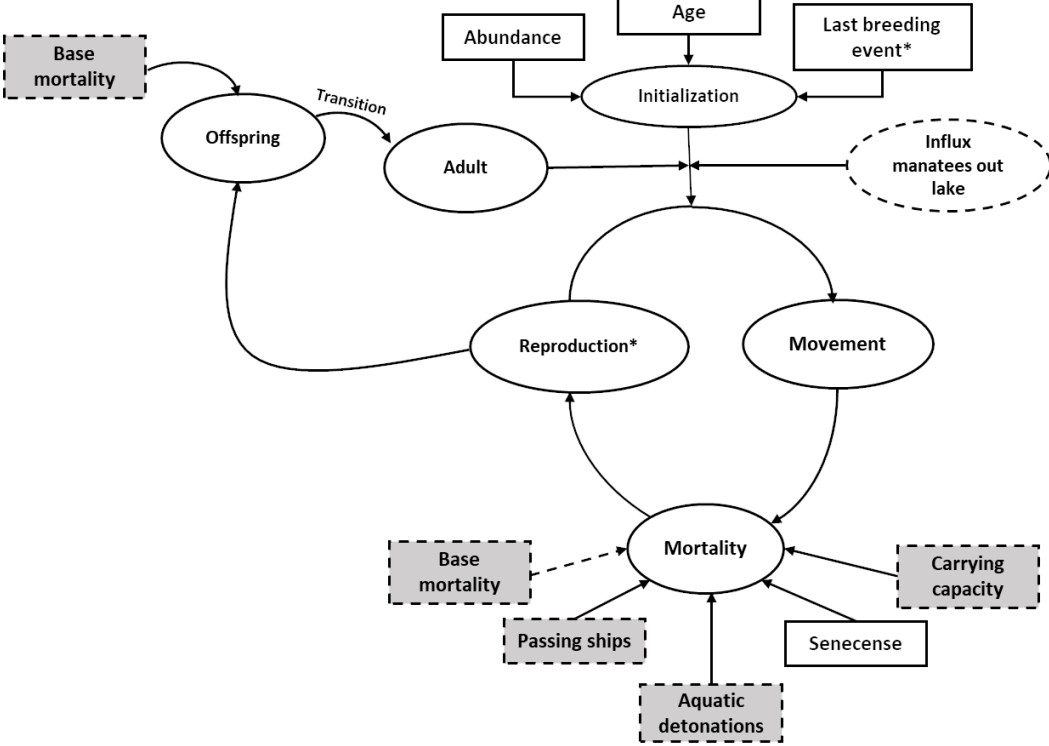

**Figure 3.** Diagram of the individual-based model (IBM). The diagram represents the sequence of the population dynamics included in IBM. Dashed lines indicate a process that can be adjusted or turned off; * indicates processes that are only part of the female breeding process. Initializaiton indicates the start or first run of the model for a particular scenario.

Carrying capacity (carrying-capacity) is the maximum number of individuals that can thrive in the lake (see point 2.3 below) when the carrying capacity is achieved, individuals (adults and offspring) start to die randomly. In addition, to represent the maximum life span of manatees, a process called senescence was added and set to a maximum lifespan of 35 years, following Marmontel et al. [25]. While it has been suggested that manatees could live up to 50 years [29], actual lifespan in the wild is much lower [25]. In addition, due to the apparent high mortality rate of manatees in the lake [10], we used a conservative estimate for manatee life span. Also, there is a 25% chance of mortality by passing ships; this process "kills" a random individual (mortality-by-ships). In addition, there is a process that simulates mortality by detonations (mortality-by-detonations) by adding a 30% chance of dying by a subaquatic explosion (Panama Canal personnel and STRI Game Wardens, [8]. A process called baseline mortality is also included, which causes a proportion of the population to die randomly. There is a different and adjustable baseline mortality set for female, male, and offspring. Following a robust review by Marmontel et al. [25] mortality was set to 0.1 (10%) for adult individuals (female and male) and 0.2 (20%) for offspring (see Appendix A for a more detailed explanation of these concepts). While there is no monitoring data available for this particular manatee population, these values of mortality are based on a study on the Florida manatee deemed a mortality rate of 10% for adults and 18% for offspring/juveniles [25]. Surveys with local communities in the Gatun area and with Canal Authority employees suggest manatee sightings are far too rare for locals to consider hunting manatees worthwhile. Therefore, hunting is not considered a significant threat to this population [10].

The reproduction process is exclusively for female manatees, and one offspring will be produced if there is at least one male manatee in the lake ("world"), and no offspring will be produced in the

absence of males. In addition, there is a minimum reproduction age for females (four years; [25,37]) and a maximum breeding percentage of the total female population (42% of the population can breed at the same time [25,38]. The newborn individuals will transition to adults in four years when they reach their fertility age (offspring-growth) [25,39]. Offspring can remain with their mothers for a period of up to 2 years, during which time a female will not reproduce [16]. However, a female can go as long as five years between calves depending on environmental stressors; for this reason, the model used an inter-calf period of 4 years [40,41].

A summary of the model parameters and the parametrization data source is presented in Table A1 in Appendix B.

### 2.3. Baseline Analysis

We ran the model 30 times over a period of 100 years to examine the population dynamics of the manatees in Lake Gatun. The model was set up with six male individuals, three female individuals, and no offspring. Mortality was set to 0.1 for adult individuals and 0.2 for offspring. Mortality by ship strikes was activated, and carrying capacity was set to 100 (a density of ~0.24 manatees/km$^2$; [42,43]). The arrival of individuals from outside the lake was set to zero. We analyzed the state variables: number of manatee individuals and age. Later we graphically analyzed these state variables. We avoided the use of inferential statistics because simulation-based analyses can be highly replicated, which can "guarantee" rejection of null hypotheses of no difference between treatments [44].

### 2.4. Sensitivity Analysis

We performed a sensitivity analysis due to the uncertainty generated by the estimates of some of the parameters and their potential effect on the model's outcomes. Four parameters and two state variables (output variables) were analyzed. The analyzed parameters were base mortality (life stage and sex), mortality-by-ships, mortality-by-detonations, manatee-influx, and carrying-capacity. The state variables included in the analysis were abundance (total number of individuals) and age (years).

The analysis was carried out in a 100-year timeframe, which was presumed to be appropriate. The sensitivity analysis was carried out using a "one-at-a-time" approach. The chosen baseline parameter values were changed either by 20% or 40%, one at a time, while keeping the rest of the parameters fixed. For each parameter change, the model was run 30 times per 100 years until all the possible parameter combinations were made. This analysis did not consider any parameter interaction. A parameter sensitivity estimation was carried out using the mean for each state variable across all the replicate runs ($n$ = 30) for each chosen parameter following [28,45].

### 2.5. Evaluation of the Long-Term Viability of the Population

We modeled four scenarios to determine under which conditions the original population would be able to be maintained over time. We explored the effect of the influx of new individuals from outside Lake Gatun (from the Caribbean coast) and the effect of mortality on the manatee population of the lake. Each scenario was replicated 30 times and run for 100 years. We designed the following scenarios:

(a) Favoring gene flow with no conservation activities: increasing the influx of manatees from the Caribbean and increasing mortality from collisions with ships due to the newly inaugurated expansion of the Panama Canal. We increased the influx of manatees and the mortality from collisions with ships by 10%.

(b) Conservation activities in place: decreasing the mortality from collisions with ships due to better management strategies and manatee conservation plans. By implementing this strategy, mortality decreased by 10%.

(c) Favoring gene flow among the Lake Gatun and the Caribbean population: we increased the number of individuals entering the lake by one individual per tick.

(d)  Favoring gene flow with conservation activities: we added one individual per tick and reduced the mortality from collisions with ships by 10%.

## 3. Results

### 3.1. Baseline Analysis Results

Simulations showed that starting with the introduction of manatees in 1964, with no influx of manatees from the Caribbean and with no conservation activities, the manatee population in Lake Gatun would plummet after 39 years. However, the timeframe during which the population becomes extinct is variable. Most of the simulations were in the range of 11–20 years (66.7%) and 21–30 years (20%) (Figure 4). Only a small proportion of the simulations were in the range of 1–10 (6.7%) and 31 to 41 (3.3%) years, respectively (Figure 4). There was a steady decline in the population, with some increase in the first 5 to 10 years (Figure 5). However, reproduction was not fast enough to maintain the population or to allow growth in the long-term. The last standing individual survival timeframe in the multiple runs corresponded to 38, 20, and 19 years for females, males, and offspring, respectively (Figure 6). Mean male age dropped dramatically, and the model shows individuals died at a young age and died very fast before they are able to reach reproductive maturity. Conversely, the mean decrease in female age is not as abrupt as with the males. The maximum mean age for females observed in some model runs was 34 years and 19 years.

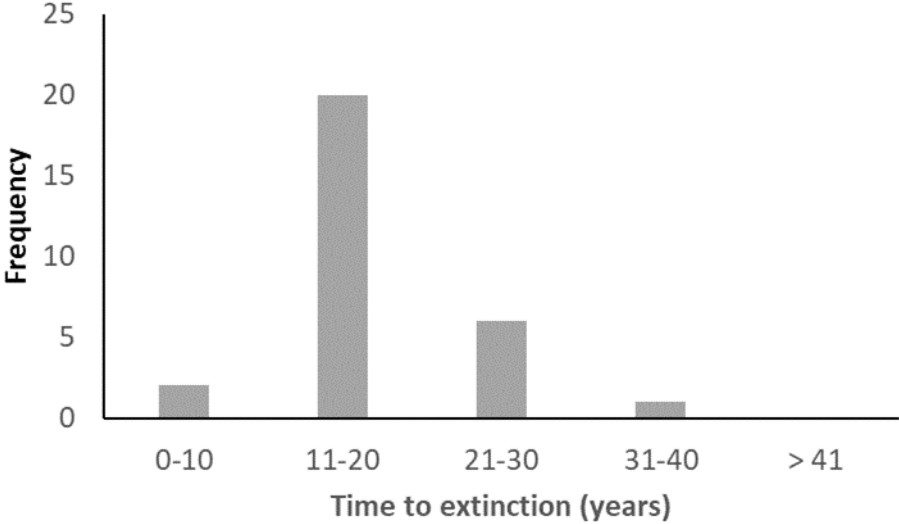

**Figure 4.** Number of simulations (*n* = 30) and the period of time (in years) where the simulated manatee population manages to survive (i.e., until the last individual disappears from the lake).

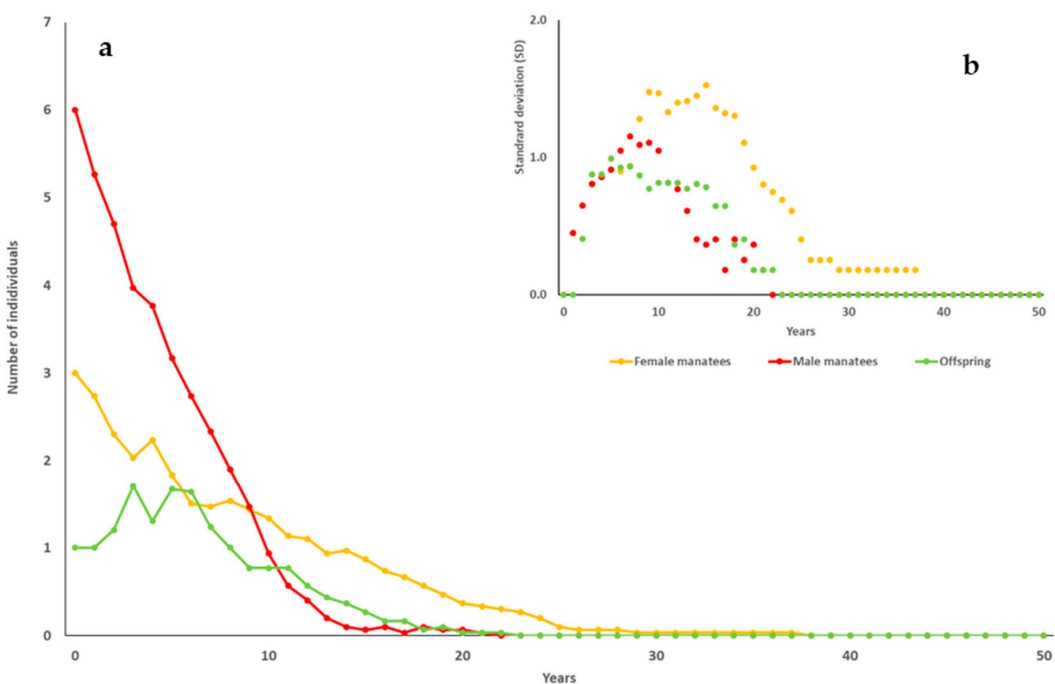

**Figure 5.** (**a**) The mean number of individuals per year for *n* = 30 runs; (**b**) the standard deviation of the mean number individuals per year for all the runs (*n* = 30).

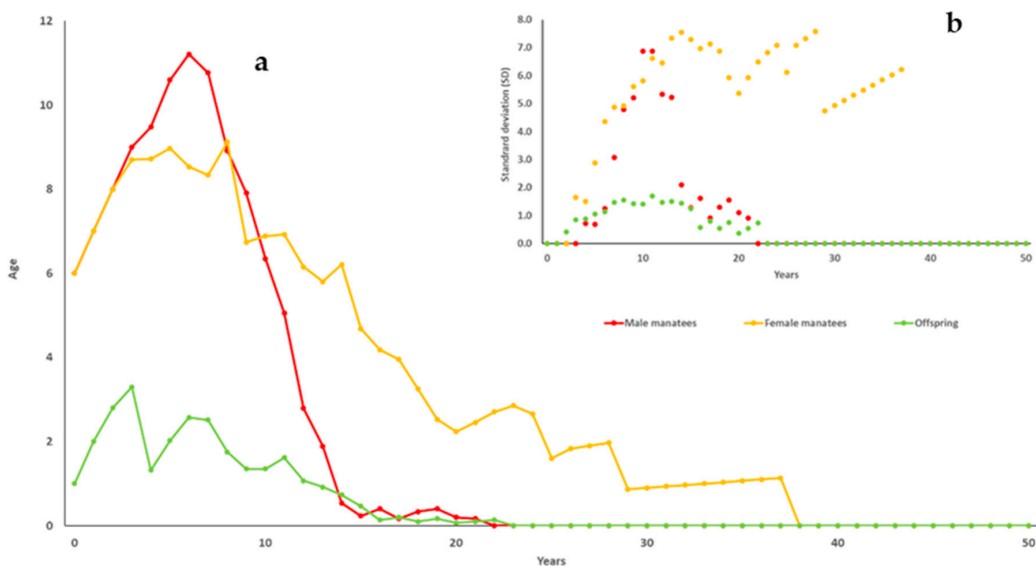

**Figure 6.** (**a**) The mean age of individuals per year for *n* = 30 runs; (**b**) the standard deviation of the mean age individuals per year for all the runs (*n* = 30).

*3.2. Sensitivity Analysis*

In general, the sensitivity analysis showed that manatee abundance and age are highly sensitive to changes in manatee influx (from the Caribbean) and offspring mortality (Tables 1 and 2). There were some differences depending on the sex and life stage of the incoming manatees. For males, in particular, the most important parameters were manatee influx and offspring mortality, although, for this latter parameter, only a decrease in offspring mortality had an effect on the abundance of males. Abundances of female manatees were sensitive to changes in offspring mortality (positive and negative), decrease in female mortality, and manatee influx (increase and decrease) (Table 1). Offspring abundance was sensitive to the influx of manatees to the lake and offspring mortality. In terms of changes in age, for all

individuals, independently of sex and life stage, an increase in the manatee influx (one-unit change) and on mortality of offspring had an effect on lifespan (Table 2).

**Table 1.** Overall ranks of the parameters that showed sensitivity to changes on ±20% for abundance per sex and life stage. Due to the nature of the manatee-influx parameter, units are the number of individuals, and changes cannot be less than zero or fractional. The "=" indicates a tie in rank.

| Parameters | Original Parameter % Variation | Parameter Values | Abundances | | | Rank of the Mean Rank |
|---|---|---|---|---|---|---|
| | | | Males | Females | Offspring | |
| Carrying capacity | −20 | 120 | 10 | 10 | 10 | 10= |
| Carrying capacity | 20 | 80 | 10 | 11 | 10 | 10= |
| manatee-influx | 2 | 2 | 5 | 5 | 5 | 5 |
| manatee-influx | 1 | 1 | 2 | 5 | 1 | 3= |
| Mortality-detonation | −20 | 0.24 | 10 | 9 | 10 | 9= |
| Mortality-detonation | 20 | 0.36 | 10 | 10 | 10 | 10= |
| Mortality_of_female | −20 | 0.08 | 7 | 5 | 5 | 6 |
| Mortality_of_female | 20 | 0.12 | 9 | 8 | 9 | 9= |
| Mortality_of_male | −20 | 0.08 | 10 | 9 | 10 | 10= |
| Mortality_of_male | 20 | 0.12 | 9 | 10 | 10 | 10= |
| Mortality_of_offspring | −20 | 0.16 | 3 | 2 | 3 | 3= |
| Mortality_of_offspring | 20 | 0.24 | 6 | 3 | 3 | 4 |
| Mortality_of_ships | −20 | 0.24 | 7 | 10 | 8 | 8 |
| Mortality_of_ships | 20 | 0.36 | 9 | 8 | 9 | 9 |

**Table 2.** Overall ranks of the parameters that showed sensitivity to changes on ±20% for age per sex and life stage. Due to the nature of the manatee-influx parameter, units are the number of individuals, and changes cannot be less than zero or fractional. The "=" indicates a tie in rank.

| Parameters | Original Paramenter % Variation | Parameter Values | Age | | | Rank of the Mean Rank |
|---|---|---|---|---|---|---|
| | | | Males | Females | Offspring | |
| Carrying capacity | −20 | 120 | 10 | 11 | 10 | 11= |
| Carrying capacity | 20 | 80 | 10 | 11 | 10 | 11= |
| manatee-influx | 2 | 2 | 6 | 13 | 14 | 11= |
| manatee-influx | 1 | 1 | 1 | 2 | 1 | 1 |
| Mortality-detonation | −20 | 0.24 | 10 | 9 | 9 | 9= |
| Mortality-detonation | 20 | 0.36 | 10 | 8 | 9 | 9= |
| Mortality_of_female | −20 | 0.08 | 7 | 6 | 5 | 6 |
| Mortality_of_female | 20 | 0.12 | 8 | 8 | 8 | 8 |
| Mortality_of_male | −20 | 0.08 | 5 | 6 | 5 | 5 |
| Mortality_of_male | 20 | 0.12 | 10 | 8 | 9 | 9= |
| Mortality_of_offspring | −20 | 0.16 | 5 | 5 | 5 | 5 |
| Mortality_of_offspring | 20 | 0.24 | 4 | 3 | 3 | 3 |
| Mortality_of_ships | −20 | 0.24 | 10 | 5 | 7 | 7 |
| Mortality_of_ships | 20 | 0.36 | 10 | 11 | 10 | 10 |

### 3.3. Evaluation of the Long-Term Viability of the Population

Results from the scenario favoring gene flow with no conservation activities (Figure 7a) showed that there was no improvement in the number of individuals in the lake in the long-term. On average, the number of individuals reached zero after 18 ± 16.2 years (Figure 7a). The scenario where only conservation activities were in place, such as decreasing manatee mortality as a result of collisions with ships or underwater detonations (Figure 7b), the survival of the population was 32.96 ± 23.9 years on average, and only 7% of the simulations reached a 100-year timeframe (Figure 7b). When we simulated gene flow (Scenario C), that is adding individuals from outside the lake (be it by human introduction or unaided influx from the Caribbean Sea), the population collapsed after 25 ± 21.9 years on average, and only one simulation managed to reach 100 years (Figure 7c). The simulations of gene flow in combination with conservation activities, such as decreasing mortality from collisions with ships or from subaquatic detonations (Scenario D), showed an average of 38.66 ± 27.23 years of population survival. Only 13% of the simulations were able to maintain a viable manatee population for the full 100-year run (Figure 7d).

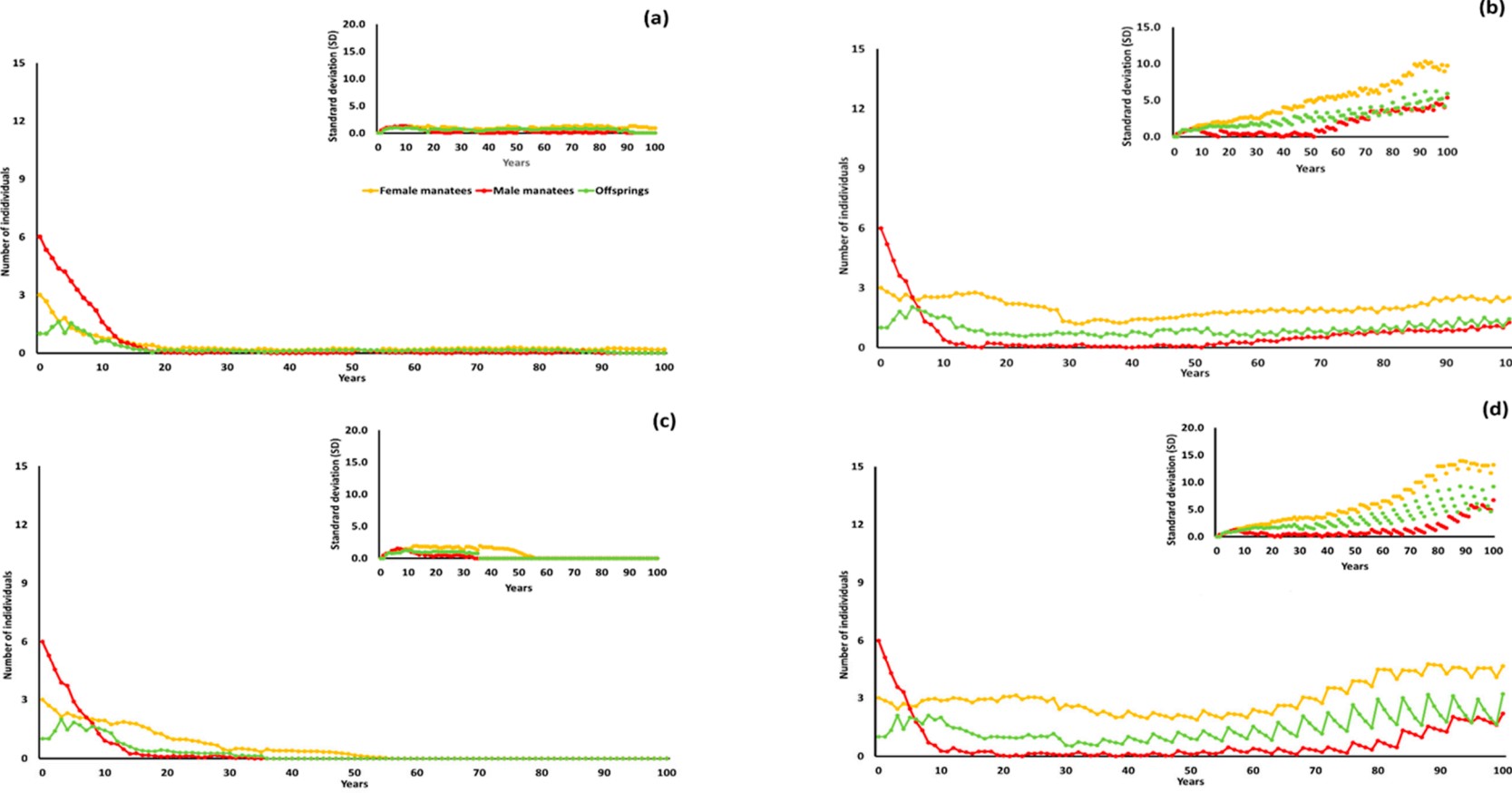

**Figure 7.** The results from different scenarios to evaluate the long-term viability of the population. (**a**) Scenario favoring gene flow with no conservation activities (Scenario A); (**b**) only conservation activities were in place (Scenario B); (**c**) gene flow (Scenario C); (**d**) gene flow in combination with conservation activities.

## 4. Discussion

In this study, we describe the use of a simple IBM model that was developed to gain insight regarding the viability of a unique population of West-Indian manatees (*T. manatus*) in Panama. Our results show that without further anthropic introductions or natural/unaided influx from the Caribbean sea, the manatee population in the lake would have collapsed quite quickly after its initial introduction in 1964. The results of our study suggest manatees are indeed managing to find a way into Lake Gatun. The likely scenario is of a lone manatee or a small group of manatees foraging in the mangroves near Gatun Locks and entering the Canal locks along with smaller vessels (e.g., sailboats) when the locks open onto the Caribbean Sea (Figure 2). Another semi-isolated manatee population in Blue Lagoon, Florida, was able to enter and depart the lake in a similar fashion when the flood control structures were open [46]. The typically murky waters in the locks would make manatees difficult to see, and they would pass undetected into the lake.

The model developed in this study provides indirect evidence of manatees entering and exiting the Canal, and suggests that the lake population is viable in the long term if management practices and conservation action are put in place. Because the West-Indian manatee is critically endangered throughout its range [1], the manatee population in Lake Gatun, although small, would be of great importance to Panama and manatee conservation in the region. In contrast to manatees in Bocas del Toro province, in Lake Gatun, we have an opportunity to develop and implement a comprehensive management plan for this species for two main reasons. First, in Bocas del Toro, manatees are found in remote rivers and coastal lagoons, where they are difficult to protect from illegal hunting and pollution and human development are much larger threats than in the Canal [9]. Second, the economic importance of the Canal and the resources that would likely be readily available, not only in terms of funding but of infrastructure and passive monitoring from canal personnel, would likely reduce the long-term costs of a conservation and management program. In addition, the presence of the Smithsonian Tropical Research Institute as scientific and technical advisers would make a management plan easier to implement, as well as to assess its success and make modifications accordingly.

While different types of ecological models to assess manatee populations exist, such as population viability analyses (PVAs) [4,25,47], these models were developed for different populations and purposes, and with a much broader scope. Our results lead to insights on this particular manatee population and add to a small but growing body of literature on the unique manatees of Panama [6,10,14,26,48,49]. However, due to the lack of data about current population demographics in Panama, we could not contrast the model's performance with real data. For this reason, the model presented in this study is limited to exploring trends in the population dynamics and was not intended as a demographic tool. However, the model is flexible enough that it can be adjusted to model other populations of manatees.

In our study, the sensitivity analysis showed that the most sensitive parameters were mortality-of-offspring and manatee-influx. For the first parameter, we have literature to support the parameter value used (e.g., [25]). For the second parameter (the number of manatees that potentially enter the lake from the Caribbean Sea), the information was scarce and mostly anecdotal (e.g., captains' logs of the Aquatic Vegetation Control Unit). Nonetheless, our model allows us to estimate the minimum population structure and size needed for long-term viability. Considering the effects of annual mortality, the manatee population in the Lake would need a minimum of 10–12 adults and 3 offspring per year. These numbers only represent an estimate since we have no control over the age or sex of the manatees that would enter the lake from the Caribbean. Given the above, our model represents an important management tool for the Panama Canal Authority. Monitoring programs would do well to ensure a minimum number of manatees in the lake and should consider re-introductions when manatee numbers drop below an established threshold.

The sensitivity analysis showed that the manatee influx parameter is crucial to maintain population viability in the lake in the long-term. More studies are needed to better estimate the number of individuals that enter into the lake each year. In addition, the simulated scenarios showed that decreasing mortality by collision with ships has a direct influence on the viability of the population in

the mid-term. When combined, manatee influx from outside the lake had no positive effect on manatee abundance when mortality by collision with ships was higher than 30%. However, in the scenario with a positive influx of manatees and a decrease in mortality by collision with ships, the abundances were not as high as the scenario with no manatee influx. Clearly, manatee influx is a key parameter for population viability. With the recent expansion of the Panama Canal (which began operations in late 2016), a larger number of much bigger ships now navigate the lake, increasing the probability of manatee mortality through collisions. However, the higher number of ships also means that the locks to the Caribbean Sea open more often, increasing the probability of manatees entering the lake.

While the Amazonian manatee (*T. inunguis*) introduced with the original population could have mated and produced hybrids [34], we considered it unlikely. Because Amazonian manatees are smaller than West-Indian manatees [29], and because West-Indian manatees typically form mating herds where several males jostle and compete for access to the female [35], it is unlikely the Amazonian manatee would have successfully competed and sired any offspring. Nonetheless, hybrids between Amazonian and West-Indian manatees occur naturally in the Amazon basin [34]. Future studies would do well to examine the genetic composition of this population to determine the degree of hybridization and the potential consequences for the long-term conservation of this population. A previous attempt to collect tissue samples for genetic analysis from the manatees in the lake proved unsuccessful [8], mainly due to the elusive nature of the manatees in the lake, and to the lack of other material (e.g., feces) that would allow for non-invasive sampling [48].

Our study is the first to use an ecological model to assess the long-term viability of the manatee population in the Panama Canal. The results of the model provide indirect evidence of manatee influx into Lake Gatun, and, as such, our results have important management implications for the long-term conservation of this population. For example, management practices could include annual population monitoring. Due to the seasonal water turbulence, sonar detection would likely be a better methodology than aerial surveys. Indeed, radar is a more effective way to estimate manatee populations in murky waters than aerial surveys and has already been used on the manatee population in Bocas del Toro with some success [6]. In addition, other initiatives could be implemented, such as sections of the lake that are identified as important manatee reproductive sites would have strict controls regarding vessel types and speed. An example of a similar measure already implemented in the lake is the case with Barro Colorado Island Natural Monument (BCI), where the speed and type of watercraft, and activities (e.g., fishing) are limited and regularly enforced. Finally, genetic studies would allow us to better understand the health of the population and design effective future management strategies.

**Author Contributions:** Conceptualization, G.M. and N.S.M.; formal analysis, N.S.M.; investigation, G.M.; methodology, G.M. and N.S.M.; writing—original draft, G.M. and N.S.M.; Writing—review and editing, G.M. and N.S.M. All authors have read and agreed to the published version of the manuscript.

**Funding:** This research was partly funded by Rufford Small Grants for Nature.

**Acknowledgments:** The authors would like to thank to George Perry for his insights and help on the early versions of this model and to the Panama Canal Authority Aquatic Vegetation Control Unit for their cooperation and support.

**Conflicts of Interest:** The authors declare no conflict of interest.

## Appendix A

*Appendix A.1. Overview*

Appendix A.1.1. Model Description

The model described below follows the overview, design concepts, and details (ODD) protocol described by Grimm et al. [33]; this approach is designed to support comprehensive and transparent

model description and communication. For that reason, some information is repeated from the general description presented in the main document.

### Appendix A.1.2. Purpose

The main objective of the model was to assess Lake Gatún's manatee population in the short- and mid-term, under the introduction of nine individuals back in the 1940s, with and without any influx from the Caribbean Sea.

### Appendix A.1.3. Entities, State Variables, and Scales

The elementary entity or unit included in the model was an individual manatee. Manatees were classified into two groups according to their life-stages: offspring (individuals younger than four years) and adults. Each offspring individual is defined by the following state variables: age and sex. Adults are characterized by age, sex, and last reproduction year (only for adult females). Each tick represented one year, and simulations were run for 50 years.

### Appendix A.1.4. Process Overview and Scheduling

The model starts every year or tick with a series of routines, as described in Figure 1. The different routines are performed in the following order: initial demography (abundances), age, last breeding event, mortality, ecosystem capacity, and manatee influx from outside the lake. Some of these routines can be adjusted before running the model by the user by using activating Boolean switches (on–off) (mortality by ships) and sliders (abundances, individuals' mortality, ecosystem capacity, and manatee influx).

### *Appendix A.2. Design Concepts*

Emergence: Changes in abundances of manatees can be observed due to the interaction among the different manatee life cycles and within their environment.

Sensing: Individuals "know" their life cycle stage (offspring or adult), their sex, their age, and the last breeding year (female manatees).

Stochasticity: Most of the model components are stochastic.

### *Appendix A.3. Details*

### Appendix A.3.1. Initialization

Individuals are randomly distributed in the lake ("world"). The initial abundances can be set by a set of sliders for offspring, female, or male. However, for the purpose of this study, the abundances were fixed to three females and six males manatees. No offspring were included at the initialization stage. The age of the initial individuals was set to six. The last breading year for females was set randomly to a value from 0 to 4. Mortality was set to 0.2 for all manatees (offspring and adults).

### Appendix A.3.2. Input Data

The model does not include any input data to model time-varying processes.

### *Appendix A.4. Submodels*

### Appendix A.4.1. Initial Demographics

The main goal is to simulate the introduction of manatees to the artificial Lake Gatún. Hence, the model is initialized with a fixed number of adult individuals (four males and six females) following [14]). Because the scenario did not include offspring, they were not included at the initialization stage.

Appendix A.4.2. Movement

Manatees can move randomly around the lake ("world") at each tick.

Appendix A.4.3. Reproduction

Reproduction is considered as a process that includes at least one male manatee present in the lake. In addition, only female manatees that are at least four years old and their last breeding being at least four years ago can reproduce. The reproduction process will produce one offspring and will reset the last breeding season to zero for the female manatee that gave birth (last-reproduction-year).

Appendix A.4.4. Mortality

Base Mortality

Annual manatee mortality follows the formula:

$$M_{base} = (Mortality_{rate} \times Individuals_{total}) \tag{A1}$$

$M_{base}$ represents the rate of the total population that will randomly die. There is a specific $M_{base}$ for offspring, female, and male manatees.

Mortality by Ships

An annual probability of 25% of a strike was set to simulate death by ship strikes in the lake. This process will kill one individual of either sex randomly.

Appendix A.4.5. Offspring Growth

Growth is simulated by changing offspring individuals from the juvenile (offspring) category to an adult category. Any offspring individual older than four years will be considered an adult, and sex will be assigned randomly with a 50% probability of been male or female.

Appendix A.4.6. Influx of Manatees

The immigration of manatee individuals to the lake is achieve by adding a new individual of random sex (50/50 chance). The age of the new individuals ($Age_{random}$) is set by a random number generated by the following equation:

$$Age_{random} = (Max_{age} - Min_{adult\ age} + 1) + Min_{adult\ age} \tag{A2}$$

where $Max_{age}$ = 35 which is the age at death by natural causes occurs; $Min_{adultage}$ = 5, which is the age where an individual is considered an adult.

Appendix A.4.7. Carrying Capacity

Carrying capacity is represented by the maximum number of individuals that can live in the lake without depleting food resources. If the maximum capacity is reached, the probability of dying increases. The probability of dying ($Prob_{dying}$) is calculated by the following equation:

$$Prob_{dying} = (Man_{Abund} - Eco_{Cap})/Man_{Abund} \tag{A3}$$

here the total abundance of manatees ($Man_{Abund}$) includes offspring and adult manatees; $Eco_{cap}$ represents the number of individuals that the lake can sustain and this ranges from 0 to infinite.

## Appendix B

**Table A1.** Parameters of the model, including a description, unit, and parametrization source.

| Parameter | Description | Unit | Reference |
|---|---|---|---|
| Initial demographics | a fixed number of adult individuals (four males and six females) | Individuals | [14] |
| Carrying capacity | A density of ~0.24 manatees/km$^2$ | Individuals | [42,43] |
| Senescence | A maximum lifespan of 35 years | Years | [25] |
| Mortality by ships | 25% chance of mortality by passing ships; this process "kills" a random individual | n/a | [8] |
| Baseline mortality | Mortality was set to 0.1 (10%) for adult individuals (female and male) and 0.2 (20%) for offspring | n/a | [25] |
| Reproduction age | Minimum reproduction age for females set to four years | Years | [25,37] |
| Offspring growth | Newborn individuals will transition to adults in four years when they reach their fertility ag | Years | [25,39,41,49] |
| Manatee influx | Set to one individual per year | Individuals | [8] |
| Mortality detonation | 30% chance of dying by a subaquatic explosion | n/a | [8] |

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
