# Peer review of "Using Ecological Modelling to Assess the Long-Term Survival of the West-Indian Manatee (Trichechus manatus) in the Panama Canal"

_water, doi:10.3390/w12051275_

Round 1

Reviewer 1 Report

Relevant article for conservation. Well written with detailed and well-organized description. However, I would like to make some small comments.

Abstract: I suggest introducing the term conservation and related it with the authors suggestion of "addition of several individuals ...", because this is a conservation strategy

Material and Methods

2.1 Study species

Avoid repeating things that were already described on the Introduction, like for instance what is written on lines 163 and 164. An alternative is to transfer the description of manatee population to the Study species.

2.2 Model description

The figure 3 needs more resolution in order to read the parameters inside dashed lines squares

  1. Results

3.3 Evaluation of the long-term viability of the population

Lines399-401: How we can interpret the presented results from the figure 7b? Some comment to lines 406-407 in relation to figure 7d

Discussion:

Lines 424-431 are not necessary, as has already been mentioned previously in Introduction

Author Response

We will like to thank for taking the time of reviewing our manuscript, specially under the current circumstances.

Relevant article for conservation. Well written with detailed and well-organized description. However, I would like to make some small comments.

 Abstract: I suggest introducing the term conservation and related it with the authors suggestion of "addition of several individuals ...", because this is a conservation strategy

The introduction was modified, and the following underlined text was added: While there are several possible scenarios, one popular hypothesis suggests that there is a low but relatively frequent influx of manatees from the Caribbean Sea into Lake Gatun via the Gatun locks (Figure 2). Indeed, a possible long-term conservation strategy for the manatees in the lake would be to carry out periodic and strategic introductions of manatees from the Caribbean coast. Nevertheless, such introductions could only be carried out as a result of long-term monitoring and conservation program, where clear priorities and methodologies were established. To date, there is no evidence on whether manatees can pass through the Canal locks unaided.

 Material and Methods

2.1 Study species

Avoid repeating things that were already described on the Introduction, like for instance what is written on lines 163 and 164. An alternative is to transfer the description of manatee population to the Study species.

            The repeated text was deleted.

 2.2 Model description

The figure 3 needs more resolution in order to read the parameters inside dashed lines squares

            We have provided a higher resolution image. In addition, we made some changes to improve the figure. We will like to note that we have provided a higher resolution image with our submission (*.eps)

  1. Results

3.3 Evaluation of the long-term viability of the population

Lines399-401: How we can interpret the presented results from the figure 7b? Some comment to lines 406-407 in relation to figure 7d

            The text and figure 7 were modified, results from scenario 2 and 4 were wrong and there were corrected accordingly. Also, we added some comments as requested by the reviewer.

The paragraph now reads: Results from the scenario favoring gene flow with no conservation activities (Figure 7a) showed that there was no improvement in the number of individuals in the lake in the long-term. On average the number of individuals reached zero after 18 ± 16.2 years (Figure 7a). The scenario where only conservation activities were in place, such as decreasing manatee mortality as a result of collisions with ships or underwater detonations (Figure 7b), the survival of the population was 32.96 ± 23.9 years on average, and only 20% of the simulations reached a 100-year timeframe (Figure 7b). When we simulated gene flow (scenario c), that is adding individuals from outside the lake (be it by human introduction or unaided influx from the Caribbean Sea), the population collapsed after 25 ± 21.9 years on average, and only one simulation managed to reach 100 years (Figure 7c). The simulations of gene flow in combination with conservation activities, such as decreasing mortality from collisions with ships or from subaquatic detonations (scenario d) showed an average of 38.66 ± 27.23 years of population survival.  Only 13% of the simulations were able to maintain a viable manatee population for the full 100-year run (Figure 7d).

Discussion:

Lines 424-431 are not necessary, as has already been mentioned previously in Introduction

The repeated text was deleted, and the text was modified as follows: In this study we describe the use of a simple IBM model that was developed to gain insight regarding the viability of a unique population of West-Indian manatees (T. manatus) in Panama. Our results show that without further anthropic introductions or natural/unaided influx from the Caribbean sea, the manatee population in the lake would’ve collapsed quite quickly after its initial introduction in 1964.

Reviewer 2 Report

Review of Muschett & Morales General comments This is an interesting study, and the whole premise, i.e., modelling whether an initially small, and possibly isolated, population of manatees can survive and reproduce over the long term, is fascinating and worthy of publication. However, although I can comment on the biological and ecological components of this study, I am not qualified to comment on the validity of the modelling used. One suggestion that I would make is that the authors include a table that clearly shows the range of parameters used in the modelling along with the data sources, so that it is easier to assess the validity, robustness and relevance of the input data. I would also recommend that a population modeller be recruited as an additional reviewer to check validity of the modelling approach, if not done so already. I wonder though if any genetic analysis could be conducted in addition to the modelling. That could make a good follow up study to examine 1) The source of immigrating manatees, 2) whether these Lake Gatun manatees comprise a genetically distinct group, 3) the degree of connectivity between these and the next nearest populations, 4) whether any species hybrids are present. Most of my specific comments are minor, but in some cases more information is required to demonstrate that the data used in the modelling is sound. Please see the attached file for specific comments and edits.

Author Response

We will like to thank you for taking the time of reviewing our manuscript, specially under the current circumstances. 

Review of Muschett & Morales General comments This is an interesting study, and the whole premise, i.e., modelling whether an initially small, and possibly isolated, population of manatees can survive and reproduce over the long term, is fascinating and worthy of publication. owever, although I can comment on the biological and ecological components of this study, I am not qualified to comment on the validity of the modelling used.

1) One suggestion that I would make is that the authors include a table that clearly shows the range of parameters used in the modelling along with the data sources, so that it is easier to assess the validity, robustness and relevance of the input data.

A new table was added to the appendix B section including the model parameters, description, unit and parametrization source.  The following text was added in t he text: A summary of the model parameters and the parametrization data source is presented in the table B1 in the appendix B section. 

2) I would also recommend that a population modeller be recruited as an additional reviewer to check validity of the modelling approach, if not done so already. I wonder though if any genetic analysis could be conducted in addition to the modelling. That could make a good follow up study to examine 1) The source of immigrating manatees, 2) whether these Lake Gatun manatees comprise a genetically distinct group, 3) the degree of connectivity between these and the next nearest populations, 4) whether any species hybrids are present.

Text was added to the discussion to address reviewer’s recommendation: A previous attempt to collect tissue samples for genetic analysis from the manatees in the lake proved unsuccessful (Muschett 2008), due mainly to the elusive nature of the manatees in the lake, and to the lack of other material (e.g. feces) that would allow for non-invasive sampling (Muschett et al. 2009).

3) Most of my specific comments are minor, but in some cases more information is required to demonstrate that the data used in the modelling is sound. Please see the attached file for specific comments and edits. 

We could not get access to the attached file that the reviewer attached as part of this review. However, if the editor can provide it, we could go through those changes too if necessary. We will like to note that the information sources for parametrization process were included in the article and the parameters were describe in detail in the text and in the ODD protocol provided in the appendix A section. We understand that parametrization sources can get lost in the text so we followed the reviewer’s suggestion of adding a table which we included in the appendix B section of the manuscript.

This manuscript is a resubmission of an earlier submission. The following is a list of the peer review reports and author responses from that submission.

Round 1

Reviewer 1 Report

This manuscript describes a simple model to assess the survival probability of the West Indian manatee population begun by the introduction of 9 individuals to Lake Gatun, Panama in 1964. The model is clear and well described, and the problems associated with very small population size are of ongoing general interest.  Manatees in particular often have small populations, and thus predictions of the stability of small populations are relevant to conservation efforts for this taxon. 

This manuscript is organized and straightforward.  The discussion is thorough and clearly addresses some of the weaknesses inherent in the model, while still making clear its value as a potential tool for conservation planning.  The practical suggestions for manatee management at this site are useful.

The focal model is based on poor data, which should be more clearly acknowledged in the methods and/or introduction, and the bases of some of the parameter estimates should be more fully explained.   For example, how was the estimate of the probability of dying due to a ship collision derived? It was set at 30%, but why?  Surely this would vary for a variety of reasons, including the level of ship traffic in the mid-1960’s.  Also, does the 30% represent the probably of death over an individual’s lifetime, or is it annual??  The values for mortality appear to be annual, though this should be made explicit, and these are appropriately grounded in data from the literature.  There is the same lack of information for why there is a 30% chance of dying by an aquatic explosion (I would assume that groups of manatees would tend to move away from areas where construction is ongoing but do not know whether there are data on this).  These two large values are reducing the probability of population persistence, and given that the focus of this paper is on just that, those values should be further elucidated. 

The addition of new individuals from outside the lake (line 148) should be clarified.  Is there only one influx, or does that influx repeat?

Clarify that offspring manatees are considered to switch to an adult annual mortality rate at 4 years of age, if that’s the case.

Edit figure captions so that they explain the figure’s meaning.  In Figure 2, the X-axis title could be more informative as well.  Also label parts of figures where present, e.g. Figure 3 should have parts aand b.

Minor comments:  There are quite a few errors of word usage and grammar, some of which I address below.

L. 47 Although there have been some notable recent attempts to estimate manatee population sizes or assess long term survival, most countries have limited data….

53.  semicolon in place of comma after ‘lake’ to fix run-on sentence.

56.  Insert ‘the’ before ‘Caribbean.’

58.  â€˜male’ rather than ‘males.’ 

            â€˜There are no data on the age of these individuals, but ….’

63-64.  Fix run-on sentence.

69-71.  I don’t follow this.  The expectation from the first part of the sentence is that the authors will continue with something like, ‘it is likely that the population is larger than these numbers suggest.  Why is the manatee population in Gatun of great importance to Panama?  Do you mean that a better knowledge of that population would be of great importance to the conservation of that species in Panama?

Can you suggest a better estimate, or is this the best available?

73-75.  This is a bit unclear – have these things been shown to affect these manatees, or manatees in other locations?   

77.  The difficulty of sighting the manatees should be taken into consideration.  

80.  Individual-based models, rather than individual based models.

82.  Do you mean ‘and then’ rather than ‘and the?’  If not, reword for clarity. 

83.  Italicize ‘A posteriori.’

97.  Use ‘weigh’ not ‘weight.’

Fig. 1  Change ‘Offsprings’ to ‘Offspring.’  Perhaps use ‘Senescence’ here and elsewhere in manuscript rather than ‘senility.’

146.  Only 1 offspring for entire population or per female?  If for the population, why?

147.  What does ‘last breeding event is set female randomly…’ mean?

149.  â€˜Outside’ in place of ‘out.’

            Why is the term ‘tick’ needed?  It makes this more difficult to understand.  I think you mean that for each influx of manatees, the model randomly adds some number of individuals to the population.

150.  Processes not process.

152-3.  Run-on sentence - replace comma with semicolon.

158.  Use ‘called’ rather than ‘call.’

165.  â€˜newborn’

170.  â€˜set up’

171.  Mortality information is repetitive as this was given above.

172.  â€˜activated’

173. ‘the following state variables:’  or ‘these state variables:’

174-176.  Check grammar.

178.  You mean ‘sensitivity’ rather than ‘sensibility.’

187.  â€˜run’ instead of ‘ran.’

193.  â€˜persist over’ instead of ‘maintain in.’

193-194.  Do you mean the effect of different levels of immigration and rates of mortality?  Otherwise the effects are clear.

273-74.  Reword for clarity.

298.  Inthe scenario where only conservation activities were in place (scenario b),the survival â€¨

301.  This currently reads ‘Individual-based model model.’

Reviewer 2 Report

I found the article interesting mainly from the ecological informatics point of view. Firstly, I must say, that using estimation modelling for such a small population brings many uncertainties which must be adequately justified and those are missing in the paper.

My comments:

Introduction. It is not clear what is the main motivation for this research. I cannot see also the research gap (basic research, management implications??). The pros and cons of the modelling related to ecological research are insufficient. Materials and Methods. Subchapter 2.1 should have been more explanatory from the species point of view and prepare the argumentation level for used parameters. Model description is very shallow and I am not able to understand how and why authors set parameters as they did. The paper is more about the model setting and verification and I expect more information on ecological informatics. I recommend extending information about scenarios (what are these scenarios good for) and reasoning parameters setting. The determination key factors influencing the results during modelling based on sensitivity analysis  itself is not defensible. At least use some error indicators. Results.  Four scenarios were mentioned in previous chapter but results do not work with them (only a few sentences are given at the of the chapter). I would expect more information about models outputs, not only a chart. This part of the article is very weak and it is almost impossible to give some statements. It must be extended. Discussion. Discussion combines information from a real discussion part and conclusion (line 338-354). But I miss more rigorous discussion not only on found results but also across existing studies (e.g. line 326 mentions a discussion on parameters but the interconnectivity with previous part of the article is missing). I recommend adding a conclusion part contemplating for instance on management implications. After reading the paper I am not sure if the authors intended the paper should be more ITor ecological piece. From the informatics point of view many parts are missing to be possible to evaluate the model setting and verify the key determinants. The ecological or zoological standpoint is inconsistent and very poor. I recommend to reconsider the main aim of the paper and follow the best practice, combining both approaches is the best way, but is difficult to keep all requirements.